Self-generated morphology in lagoon reefs

Blakeway David 1 * fathom5marineresearch@gmail.com
Hamblin Michael G. 2
1 School of Earth and Environment, University of Western Australia , Crawley , Western Australia, Australia
2 School of Mechanical and Chemical Engineering, University of Western Australia , Crawley , Western Australia, Australia
De Baets Kenneth
* Current affiliation: Fathom 5 Marine Research, Lathlain, Western Australia, Australia

Electronic publication date: 2015 May 12
Publication date: 2015
Volume: 3
Electronic Location ID: e935
Received 2014 Nov 1; Accepted 2015 Apr 15
Copyright: © 2015 Blakeway and Hamblin
Copyright year: 2015
Copyright holder: Blakeway and Hamblin
License: This is an open access article distributed under the terms of the Creative Commons Attribution License, which permits unrestricted use, distribution, reproduction and adaptation in any medium and for any purpose provided that it is properly attributed. For attribution, the original author(s), title, publication source (PeerJ) and either DOI or URL of the article must be cited.
License URL: https://creativecommons.org/licenses/by/4.0/

Keywords: Acropora, Automaton, Cellular, Coral, Geomorphology, Holocene, Houtman Abrolhos, Reticulate, Self-organised, Space-for-time

Funding: Australian Research Council Australian Postgraduate Research Award Research funding was provided by the Australian Research Council. David Blakeway and Michael Hamblin were supported by Australian Postgraduate Research Award scholarships. The funders had no role in study design, data collection and analysis, decision to publish, or preparation of the manuscript.

==============================
The three-dimensional form of a coral reef develops through interactions and feedbacks between its constituent organisms and their environment. Reef morphology therefore contains a potential wealth of ecological information, accessible if the relationships between morphology and ecology can be decoded. Traditionally, reef morphology has been attributed to external controls such as substrate topography or hydrodynamic influences. Little is known about inherent reef morphology in the absence of external control. Here we use reef growth simulations, based on observations in the cellular reefs of Western Australia’s Houtman Abrolhos Islands, to show that reef morphology is fundamentally determined by the mechanical behaviour of the reef-building organisms themselves—specifically their tendency to either remain in place or to collapse. Reef-building organisms that tend to remain in place, such as massive and encrusting corals or coralline algae, produce nodular reefs, whereas those that tend to collapse, such as branching Acropora, produce cellular reefs. The purest reef growth forms arise in sheltered lagoons dominated by a single type of reef builder, as in the branching Acropora-dominated lagoons of the Abrolhos. In these situations reef morphology can be considered a phenotype of the predominant reef building organism. The capacity to infer coral type from reef morphology can potentially be used to identify and map specific coral habitat in remotely sensed images. More generally, identifying ecological mechanisms underlying other examples of self-generated reef morphology can potentially improve our understanding of present-day reef ecology, because any ecological process capable of shaping a reef will almost invariably be an important process in real time on the living reef.

Introduction

Coral reefs are large organic structures constructed over centuries to millennia by relatively small individual organisms. The anatomy of coral reefs varies along a continuum from ‘framework’ reefs consisting primarily of coral skeletons in growth position (Lowenstam, 1950; Fagerstrom, 1987) to ‘garbage piles’ of toppled or wave-transported corals, coralgal fragments and sediment (Hubbard, Miller & Scaturo, 1990; Blanchon, Jones & Kalbfleisch, 1997). The three-dimensional form of a reef, particularly at the framework end of the continuum, is a potential repository of ecological information, because it represents a long-term integration and distillation of interactions and feedbacks between the reef-building organisms and their physical, chemical and biological environment (Roberts, Murray & Suhayda, 1975; Hopley, Smithers & Parnell, 2007; Perry, 2011). Investigating the morphological development of reefs can therefore help provide the historical context within which present-day reef ecology is embedded (Hopley, Smithers & Parnell, 2007).

The primary influences on reef morphology differ across spatial scales, from the inherent forms of reef-building organisms at the small scale to the configuration of continental shelves at the large scale. Over intermediate scales (metres to kilometres) reefs exhibit a great diversity of forms, reflecting the multitude of interacting processes affecting them. But within the diversity is a subset of globally recurring forms, indicating that there are some consistent influences governing reef morphology worldwide (Wells, 1957; Stoddart, 1969; Stoddart, 1978; Goreau, Goreau & Goreau, 1979; Blanchon, 2011; Schlager & Purkis, 2015). Traditionally, these influences have been envisaged as external controls operating at or above the scale of the morphological features, for example substrate topography (MacNeil, 1954; Purdy, 1974; Choi & Ginsburg, 1982) or the wave field (Munk & Sargent, 1954; Roberts, 1974; Storlazzi et al., 2002). While these factors are undoubtedly responsible for many aspects of reef morphology, they raise an interesting question: what would reefs look like in the absence of such external influences? This question brings the focus down to the reef-building organisms themselves. Because these organisms cumulatively become the reef, there is significant potential for behaviour and events at their scale to be expressed in reef morphology at the intermediate scale. Such ‘emergence’ of self-organised patterns from small scale processes is ubiquitous in physical and biological systems (Nicolis & Prigogine, 1977; Ball, 1999; Camazine et al., 2001). While it is recognised that coral reefs are likely to exhibit this trait (Drummond & Dugan, 1999; Mistr & Bercovici, 2003; Rietkerk & van de Koppel, 2008; Blanchon, 2011; Schlager & Purkis, 2015), it has not been directly demonstrated.

Lagoons are the most likely settings for inherent reef growth forms to arise, as they are generally flat-floored and sheltered. Several characteristic lagoon reef forms are repeated worldwide, ranging from simple mound-like patch reefs to complex cellular1 reef networks (Fig. 1; Stoddart, 1969; Hopley, 1982; Blanchon, 2011). Patch reef development can be readily visualised in terms of expansion from a nucleus, and this mode of growth has been demonstrated repeatedly, from various nuclei including topographic highs in underlying limestones (Halley et al., 1977; Mazzullo et al., 1992), sedimentary structures (Perry et al., 2012; Novak et al., 2013), or early-colonising corals (Jones, 1977; Crame, 1981). Cellular morphology, in contrast, is not an intuitive growth form. Cellular reefs distinctly resemble negative landforms, particularly karst terrains (terrestrial erosion landforms created in limestone through dissolution by rainwater). Based on this resemblance, and the recognition that the foundations of most reefs have been exposed to at least 100,000 years of weathering during Interglacial periods, cellular reef morphology has long been interpreted as an inheritance from underlying karst (Fairbridge, 1948; Purdy, 1974; Guilcher, 1988; Searle, 1994; Macintyre, Precht & Aronson, 2000; Purkis et al., 2010; Kan et al., 2015. However, there has always been an opposing view attributing cellular morphology to reef growth (Dakin, 1919; GIE Raro Moana, 1985; Collins et al., 1993; Wyrwoll et al., 2006; Barott et al., 2010; Blanchon, 2011; Schlager & Purkis, 2015). The growth alternative is gradually gaining acceptance, having been confirmed for the cellular reefs of Mataiva Atoll in French Polynesia (GIE Raro Moana, 1985; Rossfelder, 1990) and the Houtman Abrolhos Islands in Western Australia (Collins et al., 1993; Collins, Zhu & Wyrwoll, 1996; Collins, Zhu & Wyrwoll, 1998; Wyrwoll et al., 2006). Seismic surveys and coring at Mataiva showed the Holocene reef to be 10 to 20 m thick and demonstrated that, although the reef is underlain by a karst Tertiary limestone, the karst features are relatively small-scale and were infilled before submergence, such that the present reef morphology is independent of the substrate (Rossfelder, 1990). Similarly, seismic and coring in the Abrolhos lagoons recorded a Holocene reef thickness of 40 m over a flat Last Interglacial grainstone, again demonstrating independence from the substrate (Collins et al., 1993; Collins, Zhu & Wyrwoll, 1996; Collins, Zhu & Wyrwoll, 1998).

Figure 1 Cellular reefs in the Pelsaert Group lagoon, Houtman Abrolhos Islands, Western Australia (28°54′S, 114°E).

Reproduced by permission of the Western Australian Land Information Authority (Landgate) 2015.

While the seismic and coring has proven beyond reasonable doubt that the cellular reefs of Mataiva and the Abrolhos have grown into their present configuration, it has not provided a generally accepted growth mechanism. Four alternative mechanisms have been proposed. The first, developed independently by GIE Raro Moana (1985) at Mataiva and Barott et al. (2010) at Millenium Atoll, is the colonisation of the lagoon floor by networks of massive corals, which are subsequently colonised by other corals and grow upward to the surface. The second, proposed by Wyrwoll et al. (2006) for the Abrolhos, is growth to sea level of isolated branching Acropora pinnacles and stellate (star-shaped) reefs, which subsequently extend laterally and coalesce to surround enclosed depressions. The third, proposed by Blanchon (2011), is a self-limitation mechanism based on negative feedback between reef growth and water circulation—reef growth reduces water circulation which reduces reef growth, such that the cellular depressions become self-reinforcing as the surrounding reefs grow. The fourth mechanism, proposed by Schlager & Purkis (2015) is biological self-organisation through short-range support and long-range inhibition, conceptually based on Turing’s (1952) reaction–diffusion mechanism of natural pattern formation.

The alternative mechanisms outlined above are hypothetical and have not been comprehensively evaluated in real cellular reefs. In this article we use field observations and reef growth simulations to examine the process of cellular reef development in one of the type examples of cellular reefs mentioned above, those of the Houtman Abrolhos Islands. These reefs are an ideal case study site due to their flat pre-Holocene substrate, known accretion history and very pure reef-building community—cores through the Abrolhos cellular reefs consist almost entirely of branching Acropora, with a few tabular Acropora appearing as the reefs approached sea level (Collins et al., 1993; Collins, Zhu & Wyrwoll, 1996; Collins, Zhu & Wyrwoll, 1998). Furthermore, an apparent sequence of cellular reef development is evident in the Abrolhos lagoons (Wyrwoll et al., 2006), progressing from pinnacle reefs to stellate reefs surrounding semi-enclosed depressions to a reef platform surrounding enclosed depressions. Under the assumption that these are sequential stages of reef development, surveys of the pinnacle-stellate-platform sequence represent surveys through time. Space-for-time substitution (Darwin, 1842; Davis, 1899; Maxwell, 1968; Hopley, 1982) can therefore be applied to describe the evolution of the Abrolhos cellular reefs and, potentially, to reveal their formative mechanism.

Reef survey

Methods

We examined replicate sites of each stage in a 15 km2 cellular reef complex known as the Maze in the Easter Group of the Abrolhos (Fig. 2). We surveyed fifteen sites in detail and many more in brief visits, including some in the Pelsaert Group to the south and the Wallabi Group to the north. At each of the fifteen Maze sites we established four transects oriented to the cardinal directions, running upslope from the deepest to shallowest habitat. Transects varied in length from 5 m at site A (maximum depth 3 m), to 75 m at site K (maximum depth 30 m). We constructed a topographic profile of each transect by recording tide-corrected depth at one metre intervals along each transect, and quantified substrate composition by filming each transect and point counting sequential still images, using five fixed points per image (English, Wilkinson & Baker, 1997) and 25 benthic substrate categories (Data S1). The 25 substrate categories were condensed into seven categories for graphical representation: tabular Acropora, branching Acropora, massive and encrusting coral, soft coral, macroalgae, dead coral, and sediment.

Figure 2 Aerial image of the Maze in the Easter Group of the Houtman Abrolhos Islands (28°41′S 113°49′E).

The 15 survey sites are labelled A to O. Reproduced by permission of the Western Australian Land Information Authority (Landgate) 2015.

Results

Underwater observations show that the different reef stages are joined in a continuous reef blanket with a distinctive undulating form, resembling the ‘egg box’ structure described by Kan et al. (2015) in the cellular reefs of Nagura Bay, Japan. The relationship between the shapes of the different stages can be envisaged by imagining sequentially deeper horizontal slices through a solid egg box. The initial slices contact the peaks, producing circular shapes. These reefs correspond to Wyrwoll et al.’s (2006) pinnacles but we subsequently refer to them as haystacks, based on earlier descriptions of similar Acropora-dominated reefs in the Caribbean (Goreau, 1959; Kinzie, 1973). Deeper slices reach the ridges between adjacent peaks, producing stellate shapes. Subsequent slices produce a platform surrounding enclosed depressions and eventually a solid platform. Below we describe the sequence in the three idealised stages: haystack, stellate and platform. However, it should be noted that the sequence is a continuum and that sites within each stage may have features of earlier and/or later stages. Figure 3 shows representative transect profiles from each stage and Fig. 4 is a schematic block diagram incorporating the main features of the three idealised stages.

Figure 3 Representative transect profiles and benthic substrate composition graphs from haystack, stellate and enclosed sites in the Maze.

Figure 4 Block diagram illustrating the Maze’s egg-box topography and the three idealised stages of cellular reef development.

The cross-section is hypothetical but consistent with seismic and core data.

Haystacks

Haystacks occur around the margin of the Maze (e.g., sites K and M; Fig. 2). The reef surface at these sites has a sinusoidal profile, curving up over dome-shaped reef tops then descending into bowl-shaped depressions (Fig. 3). The wavelength and amplitude of the profile vary within ranges of approximately 40–100 m and 15–30 m, respectively. The Abrolhos haystacks, like those of the Caribbean (Goreau, 1959; Kinzie, 1973), consist of loosely interlocked branching Acropora colonies, most in growth position but many collapsed and overturned. Adjacent haystacks are linked by saddle-shaped ridges of branching Acropora. The predominant Acropora species on the haystack reefs and ridges are A. formosa/muricata and A. abrolhosensis. Tabular A. spicifera is abundant at site M on the exposed northern margin of the Maze but absent from the more sheltered site K to the south. Live Acropora cover is 60–100% on the reef tops and ridges at both sites, decreasing to approximately 30% within the site M depressions and 0% within the more enclosed and restricted site K depressions. Dead Acropora branches at depth are occupied by macroalgae at both sites, predominantly Sargassum spp. at site M and Lobophora variegata at site K. Beyond the outermost haystacks, the reef surface slopes down to a flat sandy seafloor at 35 m without breaking into isolated patches (Fig. 4). Corals on these outer slopes are predominantly branching and tabular Acropora to approximately 25 m depth (Wilson & Marsh, 1979). Below 25 m the coral community is more diverse, with a high proportion of foliose genera, particularly Leptoseris and Pachyseris.

Three islands of storm-deposited Acropora rubble line the eastern margin of the Maze (Collins, Zhao & Freeman, 2006), and a series of submerged east–west trending linear reef banks occur on the northern margin of the Maze. Two banks can be seen in Fig. 2 and two deeper banks lie beyond them. The bank crests are 2 to 15 m deep, sloping downward to U-shaped troughs at 20 to 30 m. The outermost bank reaches the seafloor at 35 m. Coral cover and zonation on the banks is equivalent to that of the haystack reefs.

Stellate reefs

In the stellate stage the haystack reef tops and ridges reach sea level, producing a network of flat-topped star-shaped reefs (sites B, E, F, J, L, N). Water circulation within the intervening depressions is further reduced and the water column is often stratified and stagnant. Live coral is consequently restricted to shallow depths, often less than 15 m in the more enclosed depressions. The shallow subtidal reef slopes and ridges remain dominated by live branching Acropora (Fig. 5) and occasional foliose Montipora. Dead Acropora branches at depth are colonised by Nephthea soft corals and Lobophora variegata, and fine sediment accumulates in the bases of the depressions.

Figure 5 Dense in-situ (A) and collapsed (B) Acropora colonies on a stellate reef in the Pelsaert Group lagoon.

A distinct shallow coral community begins to appear in the stellate reefs, consisting of diverse massive and encrusting corals, the most abundant genera being Montipora, Goniastrea, Favia, Favites, Merulina, Astreopora, Montastrea, Mycedium, Echinophyllia, Cyphastrea, Alveopora and Lobophyllia. Several apparent developmental stages of this community are present, initiating as a discontinuous cover of small colonies on dead Acropora branches (Fig. 6A) and culminating in vertical or overhanging walls descending from the surface to as deep as ten metres, but typically between two and eight metres (Fig. 6B).

Figure 6 Inferred early (A) and late (B) stages of reef wall development in the Pelsaert Group lagoon.

The walls appear to initiate through the colonisation of dead Acropora branches by massive and encrusting corals (A), and subsequently grow to become vertical or overhanging (B).

Reef platform with enclosed depressions

In the platform stage (sites C, D, I, O) the trends in water quality and coral distribution that were established in the stellate reefs develop further: most depressions are rimmed by vertical walls of massive and encrusting coral, live Acropora cover below the walls declines rapidly with depth, the water column is usually stratified, and the depressions typically have a deep sediment fill. Late-stage enclosed depressions (A, G, H) gradually fill with sediment to the level of the surrounding reef flat. As they fill, the fringe of live Acropora beneath the vertical walls migrates upward, eventually overgrowing the walls and encroaching over the depression floors.

Model

Rationale

Based on the survey results described above, the Abrolhos cellular reefs appear to exhibit a straightforward developmental sequence. However, they provide little direct insight into the origin of cellular morphology because the cellular form is already present at the haystack stage. Given the high coral cover on the haystack reef tops and ridges, subsequent growth will inevitably enclose the depressions. Although the haystacks must have progressed through earlier stages to reach their present configuration, the existing space-for-time sequence does not extend back to the earlier stages, presumably because coral colonisation of the Last Interglacial surface ceased when it became deeply submerged and covered by sand in the mid Holocene. If this is the case, even the youngest haystacks probably initiated more than 4,000 years ago. Several processes appear to be suppressing live coral cover, and therefore accretion, within the present-day haystack reef depressions, including reduced water circulation and macroalgal colonisation. But in the absence of earlier stages in the space-for-time sequence it is impossible to determine whether these processes could have initiated the depressions or whether they are consequential. In this situation, computer simulation provides a potential means of investigating the early stages of reef development.

Methods

The model we describe below is configured as a cellular automaton: an array of identically-programmed interacting cells (Von Neumann, 1951; Ulam, 1962; Data S2). This structure is well-suited to simulating reef growth because each cell in the array can be considered to represent a square metre of seafloor, and reefs can grow on the seafloor as three-dimensional stacks of cubic ‘corals.’ Using this approach, we simulate lagoon reef development as the growth and coalescence of patch reefs from individual coral recruits on a flat seafloor. We first describe a basic model in which colonisation and growth is essentially random and unconstrained except by sea level, and subsequently introduce a parameter representing branching Acropora.

Basic model

The basic model is initialised by defining the seafloor depth, the array dimensions and the number of coral recruits. We used a default configuration of 30 m depth (static, i.e., no sea level variation), a 160 × 160 cell array (representing 160 × 160 m, or 2.56 hectares, of seafloor), and 64 randomly-spaced coral recruits, occupying 0.25% of the array. The 160 × 160 m recruitment array was centered within a larger array of 250 × 250 m, giving the reefs a 45 m margin for lateral growth. We chose 30 m as the default depth, rather than the 40 m of the Maze, because cellular reefs elsewhere appear to be thinner than the Maze; those at Mataiva, for example, are only 10 to 20 m thick (Rossfelder, 1990). The horizontal dimensions of the array were selected to minimise computation time while still allowing adequate spatial representation of reef morphology. The colonisation density was selected such that the resulting patch reefs were close enough to eventually coalesce but not so close as to immediately coalesce. We examined variations to the default configuration, including sea level rise, and describe them later under ‘additional modifications.’

Growth from the initial coral recruits is effected by assigning two growth probabilities to each cell in the array in each iteration: a vertical probability representing the likelihood of the cell growing upward itself and a neighbour probability representing the likelihood of the cell being overgrown by a neighbouring coral (Fig. 7). The vertical probability of vacant seafloor cells is zero, and the vertical probability of coral-filled cells is random. The neighbour probability of each cell is the product of a random number between zero and one and a ‘neighbour value’ that depends on the state of the eight surrounding cells. Cells with no shallower neighbours are assigned a neighbour value of zero; otherwise, the cell’s neighbour value rises incrementally for each shallower neighbour. If a cell becomes surrounded by shallower neighbours, it is guaranteed to be overgrown. Otherwise, growth is determined by comparing the cell’s vertical and neighbour probabilities against two random numbers between zero and one. If either or both probabilities exceed their respective random numbers, the cell grows by one metre when the array is updated prior to the next iteration. Vertical accretion is halted at sea level but lateral accretion continues. The time represented by each iteration is arbitrary but we consider it to be 100 years, giving a mean vertical reef accretion rate of 7 mm/yr (the theoretical maximum rate of 10 mm/yr is not achieved because corals do not grow in every iteration).

Figure 7 Cross-section through a hypothetical model reef.

Upward-pointing arrows indicate vertical growth directions, horizontal and diagonal arrows indicate neighbour growth directions.

Branching Acropora model

Representation of branching Acropora was guided by the output of the basic model (Fig. 8). The basic model reefs have an irregular ‘spiky’ surface, with corals projecting up to four metres above the surrounding reef. Such projections cannot occur on real branching Acropora reefs because, due to their ‘brittle tree’ morphology, any branching Acropora colonies that grow more than a metre or two above their surroundings will inevitably collapse (Maragos, 1972; Bak, 1976; see Fig. 5B). This is not necessarily a disadvantage. Because broken fragments can survive and grow to form new colonies, collapse and fragmentation are recognised as inherent and significant modes of reproduction and short range dispersal in branching Acropora (Gilmore & Hall, 1976; Tunnicliffe, 1981; Bothwell, 1982; Highsmith, 1982). Collapse is represented in the branching Acropora model by imposing a maximum height differential between neighbours (hereafter termed collapse limit) of two metres, such that corals growing to project two metres above any neighbouring cell are prevented from growing upward until the deeper cell grows. Although they cannot grow upward, projecting corals contribute to the growth probability of neighbouring cells in two ways: first, they ‘support’ neighbouring corals, ensuring they are unrestricted by the collapse limit; and second, they may ‘collapse into’ deeper neighbouring cells (i.e., they increase the neighbour probability of those cells). Although this representation of collapse involves no subtraction of height from the projecting colony, it remains valid because it is equivalent to the projecting colony growing a metre then collapsing back a metre during the iteration. Because the imposition of the collapse limit slows reef growth, the time represented by each iteration is reduced to 40 years. This gives a mean vertical reef accretion rate of 9 mm/yr, approximating that of the Abrolhos cellular reefs (Eisenhauer et al., 1993; Collins et al., 1993).

Figure 8 Reefs generated by the basic model.

(A) Two-dimensional plan view of a patch reef after 80 iterations (8,000 years) of growth from a single seed coral. Shading corresponds to depth—the reef top at sea level is white and the surrounding seafloor at 30 m depth is black. This patch reef reached sea level in approximately 45 iterations (4,500 years), and by 80 iterations has developed a 15 m wide reef flat. (B) Three-dimensional oblique view of the patch reef in (A), showing the irregular surface morphology caused by projecting corals. The reef slopes are approximately 65°. (C) Two-dimensional plan view of a coalescing patch reef system after 80 iterations. Only the uppermost 10 m of the reef system is shown, simulating an aerial view with 10 m water visibility. (D) Three-dimensional oblique view of the reefs in C.

Additional modifications

We examined the effects of increasing the collapse limit, altering water depth, altering colonisation density, and periodically adding new coral recruits. We also simulated sea level rise and depth-dependent growth, using a simplified linear sea level rise of 10 mm/yr, stabilising at 30 m depth, and a simplified linear reduction of the coral growth rate to 10% of the surface rate at 30 m.

Results

Basic model

Patch reefs created with the basic model appear approximately circular in plan view and steeply conical in oblique view (Figs. 8A and 8B). The individual patch reefs maintain their conical form as they enlarge and coalesce with neighbouring patches (Figs. 8C and 8D). We use the term ‘nodular’ to describe the shapes and forms generated by the basic model. While the nodular reefs resemble many natural patch reefs (e.g., Fig. 9), they bear little resemblance to cellular reefs. In fact, their shapes are the inverse of cellular reefs; nodular reefs appear convex and subcircular in plan view, whereas cellular reefs are concave and stellate, surrounding subcircular depressions. However, the basic model is generic and does not intentionally represent any particular coral type.

Figure 9 Coalescing nodular patch reefs exposed on a low spring tide at Cockatoo Island in the Buccaneer Archipelago, Western Australia (16°4.8′S 123°35′E).

Photograph by John MacFadyen.

Branching Acropora model

Reefs created with the branching Acropora model closely resemble the Abrolhos cellular reefs (Fig. 10). The model reproduces the characteristic egg box form of the real reefs and all its corollaries including haystack reefs, stellate reefs with radiating ridges, reef platforms enclosing bowl-shaped depressions, scalloped platform margins and the presence of multiple small depressions within larger multi-lobed depressions. The 45° slopes of the model reefs are steeper than the mean of the real Acropora slopes (37° ± SD 6°) but within their recorded range. Figure 11 shows sequential stages in the development of the branching Acropora reefs, demonstrating the emergence of their egg box morphology. The key process is the formation of ridges between adjacent patch reefs. This process begins when the patch reefs meet, whereupon the valleys between them grow rapidly upward to become saddle-shaped ridges (Figs. 11C and 11D). The depressions surrounded by the reefs and ridges are initially irregular in outline but are progressively smoothed to subcircular shapes as the surrounding reef grows. Eventually the depressions become completely enclosed within the reef platform and infilled by coral (Fig. 11E).

Figure 10 Plan view of model (A) and real (B) branching Acropora reefs.

Only the uppermost 10 m of the model reef is shown, simulating an aerial view with 10 m water visibility. This reef grew in 100 iterations (4,000 years) from 225 corals seeded at the default colonisation density (0.25%) in a 300 × 300 m array. The real reefs shown in B are those from Fig. 1, in the Pelsaert Group of the Abrolhos. The arrowed annotations indicate the locations of photographs shown in Figs. 5A and 5B. Figure 10B is reproduced by permission of the Western Australian Land Information Authority (Landgate) 2015.

Figure 11 Sequential stages in the development of model branching Acropora reefs.

(A) Plan view of the 64 randomly spaced seed corals from which the model reefs developed. This seed coral configuration is the same as that used to create the basic model reefs in Figs. 8C and 8D. (B) After 25 iterations (1,000 years) the seed corals have developed into conical patch reefs with 45° slopes. (C) After 50 iterations (2,000 years) the patch reefs have enlarged and many have merged. When patch reefs meet, the valleys between them grow upward rapidly to become saddle-shaped ridges. (D) After 100 iterations (4,000 years) most of the reef tops have reached sea level and the system of ridges has developed to enclose and isolate depressions, producing egg-box morphology. (E) After 140 iterations (5,600 years) an extensive sea level platform has developed, and most of the depressions have filled. (F) Plan view showing the uppermost 10 m of the reef in D, simulating an aerial view with 10 m water visibility. The video shows model branching Acropora reef development at five iteration intervals to 100 iterations, then 120 and 140 iterations.

Additional modifications

Increasing the collapse limit was the most influential of the additional modifications. Progressively increasing the collapse limit beyond the two metres of the branching Acropora model produces a transition from cellular to nodular reef forms. A three metre collapse limit creates reefs with weakly developed subcircular depressions (Figs. 12A and 12B) and a four metre collapse limit creates reefs with very few depressions (Figs. 12C and 12D). Collapse limits of more than four metres produce nodular reefs equivalent to those of the basic model.

Figure 12 Influence of the collapse limit on model reef morphology.

(A) Plan view, 0–10 m depth, of a model reef system with a 3 m collapse limit after 90 growth iterations. (B) Oblique view of the model reef in A. The reef slopes are approximately 55°. (C) Plan view, 0–10 m depth, of a model reef system with a 4 m collapse limit after 85 growth iterations. (D) Oblique view of the model reef in C. The reef slopes are approximately 60°.

Varying water depth also significantly influences reef morphology. Reducing depth reduces reef thickness, which constrains the morphological expression of the collapse limit such that the appearance of the branching Acropora reefs transforms from cellular to nodular as depth decreases (Figs. 13A and 13B). In the extreme case of reefs growing in only one or two metres water depth, where the collapse limit has no effect, all variants of the model produce identical nodular reefs. Increasing depth, by itself, has little influence on reef morphology (Figs. 13C and 13D). However, more realistic representations incorporating sea level rise and depth-dependent growth cause reef slopes to steepen significantly as depth increases (Figs. 13E and 13F). Variations in colonisation density and timing have relatively little effect on reef morphology, besides the expected crowding of patch reefs at high density (Fig. 14).

Figure 13 Influence of water depth on model reef morphology.

These modifications were undertaken in a smaller (90 × 90 m) array, but maintained the default 0.25% colonisation density. (A) Branching Acropora reefs grown in 10 m depth exhibit a transition toward the nodular forms of the basic model reefs shown in (B). (C) Branching Acropora reefs grown in 50 m depth retain their cellular morphology. (D) Basic model reefs grown in 50 m depth retain their nodular morphology. (E) Branching Acropora reefs incorporating sea level rise and depth-dependent growth steepen to 60°. The blocky appearance of these reefs is a consequence of being forced to their maximum slope, which overrides the model’s randomness. (F) Basic model reefs incorporating sea level rise and depth-dependent growth steepen to approximately 85°.

Figure 14 Influence of colonisation density and timing on model reef morphology.

These plan view images show the effects of decreasing the recruitment rate from the default 0.25% to 0.125% (A: branching Acropora reef, 100 iterations), (B: basic reef, 80 iterations), increasing the recruitment rateto 1% (C: branching Acropora reef, 90 iterations, D: basic reef, 60 iterations), and periodically adding new recruits during reef growth (E: branching Acropora reef, 100 iterations, F: basic reef, 70 iterations).

Discussion

Model

The resemblance in shape and form between the model reefs and real reefs suggests that the model adequately represents reality. This interpretation is supported by the model’s simplicity: it has only one rule—collapse if too steep—which is intuitively reasonable and supported by field observations. Model reef morphology is hyper-sensitive to that rule, running through a nodular to cellular spectrum as permissible steepness is reduced and collapse becomes more frequent.

The nodular reefs produced by the basic model appear straightforward and visually ‘correct’ as growth forms, because the individual patch reefs maintain their forms as they grow and merge. This straightforward morphology is indicative of pure in situ (in place) growth. Cellular reefs are more complex because the patch reefs transform as they merge, eventually becoming linked by a network of ridges. This transformation results from the high frequency of collapse in the branching Acropora model. However, it is not simply the frequency of collapse that produces ridges; more important is the distribution of collapse. Because the valleys between merging patch reefs are low points in the reef structure, coral colonies in the valleys are less likely to project above their neighbours than corals elsewhere. Consequently, they are relatively unrestricted by the collapse limit and are therefore more likely to remain in place as they grow, and less likely to collapse, than colonies elsewhere (Fig. 15). The retention of in-situ colonies transforms the V-shaped valleys into saddle-shaped ridges that grow to sea level, enclosing depressions (Figs. 11C and 11D, Fig. 11 Video). The subcircular shapes of the depressions arise through the same non-uniform distribution of collapse. Colonies in the re-entrant concavities of early-stage depressions are supported by neighbours and therefore tend to remain in place while those on projecting convexities tend to collapse. Over time this creates smooth rounded shapes, the ultimate smooth shape being a circle.

Figure 15 Diagram illustrating the proposed mechanism of ridge formation derived from the branching Acropora model.

The diagram shows a cross-section through two merging patch reefs after 50 iterations (the two uppermost patch reefs in Fig. 11C). Isochrons at 20 and 40 iterations show that the patch reefs were initially conical, and that the valley between them has accreted rapidly since the patch reefs merged. Rapid accretion is attributed to the tendency for colonies in valleys to remain in place and for colonies on reef slopes to collapse.

Abrolhos cellular reefs

The foregoing descriptions of the model branching Acropora reefs provide two testable predictions regarding real cellular reefs. First, their slopes should have consistent and relatively low gradients, representing the angle of repose (maximum slope stability angle) of branching Acropora. Second, the proportion of in-situ colonies should be highest in the valleys and ridges between adjacent patch reefs, and lowest on reef slopes. Both predictions are supported in the Abrolhos, where Acropora slopes average 37° ± SD 6° (Fig. 3) and Acropora colonies in valleys and ridges are generally upright and in situ (Fig. 5A, see Fig. 10 for photo location) while those on reef slopes are often overturned (Fig. 5B, see Fig. 10 for photo location). We conclude that the Abrolhos cellular reefs have developed according to the model and that Fig. 11 closely describes their morphological progression.

One significant difference between the real and model reefs is the reduced accretion rate of the real reefs once they reach sea level. Model branching Acropora reefs reach sea level from 30 m depth in approximately 90 iterations (3,600 years) and only require 70 more iterations (2,800 years) to completely fill the platform, whereas the Maze reefs, in 40 m depth, reached present sea level in approximately 4,500 years (Eisenhauer et al., 1993; Collins et al., 1993) but still have not filled the platform nearly 7,000 years later. The reduced accretion of the real reefs probably results from two factors not represented in the model. The first is the reduction of Acropora cover and vitality at depth, as water circulation is restricted by the enclosure of the depressions (Wyrwoll et al., 2006). This is an example of self-limitation through negative feedback between reef growth and water circulation (Blanchon, 2011). Self-limitation is therefore a significant influence on the Abrolhos cellular reefs, but operates primarily on their accretion rate not their morphology. The second factor is the colonisation of upper reef slopes by relatively slow-growing massive and encrusting corals. The steep walls created by these corals effectively exclude branching Acropora, because any branching Acropora that colonise the walls are likely to break off once they grow too large to be supported at their base. By ‘engineering’ steep walls (sensu Jones, Lawton & Shachak, 1994), massive and encrusting corals are able to monopolise—for thousands of years—prime shallow subtidal habitat that would otherwise be occupied by fast-growing branching Acropora. The combination of reduced water circulation at depth and steep walls in the shallows restricts live Acropora to a fraction of their previous distribution, significantly slowing the overall reef accretion rate. Model cellular reefs, in contrast, rapidly fill the platform because ‘live’ Acropora occupy all subtidal habitats including the depression slopes and floors.

Another significant difference between the real and model reefs is the series of linear reef banks on the northern margin of the Maze. We interpret these as early to mid Holocene wave-deposited structures, resulting either from storms, cyclones (Scheffers et al., 2012) or tsunamis (Scheffers et al., 2008).

Application to other reefs

The morphology of cellular reefs elsewhere appears similar enough to the Abrolhos reefs to suggest they have developed the same way, an inference supported by the abundance of branching Acropora in documented examples (Alacran Reef: Hoskin, 1963; Solomon Islands: Morton, 1974; Tétembia Reef: de Vel & Bour, 1990; Cocos Keeling Atoll: Williams, 1994; Pelican Keys: Aronson, Precht & Macintyre, 1998; Elizabeth Reef: Woodroffe et al., 2004; Pohnpei: Turak & DeVantier, 2005; Tun Sakaran: Montagne et al., 2013; Nagura Bay: Kan et al., 2015). At least two of these examples, the Solomon Islands (Morton, 1974) and Nagura Bay (Kan et al., 2015), exhibit vertical walls of massive and encrusting corals above the Acropora zone, suggesting they have undergone the late-stage shallow coral community succession observed in the Abrolhos.

We have separated Mataiva Atoll from the list above as Porites is abundant there and has been considered responsible for the cellular morphology (GIE Raro Moana, 1985; Delesalle, 1985). However, branching Acropora are also abundant at Mataiva (Delesalle, 1985; Rossfelder, 1990). We suggest that branching Acropora are the primary reef builders at Mataiva and Porites are colonisers of the Acropora reef, not framework builders. Another possible exception to the rule of Acropora dominance is the ‘Type-1’ reticulate reef of the Red Sea (Purkis et al., 2010), where Porites is also abundant (Bruckner, 2011). However, we would not classify all Type-1 Red Sea reefs as cellular because, although deep, the depressions are not always circular; more often resembling the transitional depressions produced by intermediate collapse limits (see Purkis et al.’s Fig. 2). Some Red Sea reefs are distinctly cellular and we predict those to be Acropora-dominated (e.g., 27°57′N, 35°13′E). Closer examination of these and other cellular reefs is required to determine whether the predominance of branching Acropora is universal, and whether the reef slope gradients and the distribution of collapsed colonies conform to the Abrolhos example.

We have not classified the previously-mentioned Millenium Atoll with the cellular reefs listed above because the scale of the cellular morphology at Millenium and in many other Pacific atoll lagoons is up to an order of magnitude larger than the Abrolhos. While the large-scale cellular reefs also seem to consist predominantly of branching Acropora (Roy & Smith, 1971; Grovhoug & Henderson, 1976; Valencia, 1977; Barott et al., 2010), we do not believe our model applies directly to them because it cannot produce cells of their horizontal dimensions unless it is scaled up massively, to unrealistic depths of at least 100 m. We are currently investigating the large-scale cellular morphology.

The transitional and nodular shapes produced by increasing the collapse limit in the model also occur in real reefs (e.g., Fig. 9). The simplest interpretation of these shapes is that they indicate coral types, or mixtures of coral types, progressively less prone to collapse than branching Acropora. In this interpretation, transitional shapes represent reef builders that occasionally collapse, such as foliose and tabular corals, and nodular shapes represent reef builders that rarely collapse, such as massive and encrusting corals or coralline algae. The nodular reefs of Cockatoo Island in Fig. 9 conform to this interpretation, as they consist of massive and encrusting corals cemented by coralline algae (D Blakeway, pers. obs., 2010). However, the model indicates that transitional and nodular reef shapes are not necessarily diagnostic of coral type, because branching Acropora patch reefs appear nodular (i.e., circular in plan view) before they merge with adjacent patch reefs, and transitional to nodular after they merge in shallow water (e.g. Fig. 13A). This suggests that additional information on water depth, reef thickness and reef slope gradients will be required to reliably infer coral type from reef morphology in transitional and nodular reefs. Such three-dimensional data are becoming increasingly available through reef-oriented remote sensing (Zawada & Brock, 2009; Zieger, Stieglitz & Kininmonth, 2009; Goodman, Purkis & Phinn, 2013 and references therein; Leon et al., 2013; Leon et al., 2015). In two-dimensional aerial images, however, the only diagnostic morphology is cellular—signifying relatively thick (>∼10 m) reefs constructed by collapse-prone organisms.

Branching Acropora

The default collapse-prone reef builders on modern reefs are branching Acropora. While it seems possible for other branching coral genera or other calcareous branching invertebrates (e.g. Millepora) to create cellular reefs, observations worldwide (listed above) suggest it is almost exclusively Acropora: A. cervicornis in the Atlantic and multiple species in the Indo-Pacific. This is probably because branching Acropora have the ultimate strategy for rapid pre-emption of space in lagoon environments. Acropora branches not only grow quickly (up to 19 cm/yr in the Maze; Blakeway, 2000), they regularly develop new branches which themselves branch and rebranch, giving them the potential for exponential expansion (Shinn, 1976). Constant growth, branching and collapse produce an open three-dimensional structure that rapidly fills lagoons (Davis, 1982). Our model indicates that, given adequate depth, cellular reefs are the inevitable result. Cellular reefs are essentially a phenotype of the branching Acropora genome(s), emerging from the innate behaviour of branching Acropora colonies just as colony morphology emerges from the innate behaviour of polyps.

If the relationship between cellular reef morphology and branching Acropora holds, the distinctive shapes of cellular reefs in remotely sensed images can potentially be used to identify and map branching Acropora habitat. This could be useful in reef conservation, as the sensitivity of branching Acropora to environmental conditions makes them something of a canary in the coral reef coalmine (Marshall & Baird, 2000; Loya et al., 2001; Acropora Biological Review Team, 2005; Roth & Deheyn, 2013). However, assessing anthropogenic impacts in apparently degraded Acropora-dominated lagoons will rarely be straightforward, because natural self-limitation and community succession can drastically reduce Acropora cover and vitality in the mid to late stages of reef development. Aronson, Precht & Macintyre (1998), Aronson (2011) and Perry & Smithers (2011) highlight the value of documenting and understanding such intrinsic trends, generated by the reef itself, before attempting to evaluate the effects of extrinsic influences imposed from outside the reef, including anthropogenic stresses.

Conclusions

Our simulations indicate that reef morphology is fundamentally determined by the extent to which reef-building organisms either remain in place or collapse. This control is best expressed in lagoons, where diminished hydrodynamic and substrate influences allow reefs to grow into their inherent forms. The purest growth forms arise in sheltered lagoons dominated by a single type of reef builder, as in the cellular reefs of the Abrolhos. In these situations, reef morphology can be considered a phenotype of the predominant reef-building organism.

While the propensity for collapse appears to explain the nodular to cellular spectrum of lagoon reef morphology, many more relationships between reef ecology and morphology remain to be discovered. Many of the recurrent patterns in reef morphology are likely to be ecological phenomena (Blanchon, 2011; Schlager & Purkis, 2015). Quantifying these patterns and identifying their underlying mechanisms can potentially improve our understanding of present-day reef ecology, because any ecological process capable of shaping a reef will almost invariably be an important process in real time on the living reef.

Investigation of the relationships between reef morphology and ecology is benefiting from advances in the availability, resolution and processing of remotely sensed imagery. However, the single most important research technique remains careful and objective underwater observation. Any consistent correlations between reef morphology and underwater survey data, such as coral type, can be considered potential causal relationships warranting closer examination.

In surveying modern reefs, it should be recognised that a reef’s current state may not represent its formative state, particularly if the reef has reached sea level. While seismic and coring can access the history stored within such reefs, both techniques are logistically demanding and expensive. The complementary methods we employed in the Abrolhos, space-for-time substitution and computer simulation, are relatively simple and inexpensive but can provide a comprehensive reconstruction of a reef’s history and a sound basis for extrapolating its future development.

Supplemental Information

Data S1 Benthic substrate point count data from the 15 Maze sites

Click here for additional data file.

Data S2 Model code

Click here for additional data file.

Many thanks to field volunteers Nik Sander, Gary Watson, Tim Benfield, Nokome Bentley, Matz Berggren, Freda Blakeway, Richard Campbell, Robin Cornish, Simon Cutler, Geoff Deacon, Tim Fisher, Steven Fletcher, Erez Jacobsen, Margaret Jones, Jason Kennington, Helen Kirby, Clare Taylor and Roland Tyson. Logistic support from the mainland was provided by Freda Blakeway, Philip Blakeway and Wendy Perriam. Logistic support in the Islands was provided by the Western Australian Department of Fisheries, especially Kim Nardi, Michael Byrne, Chris Chubb , Boze Hancock, David Murphy and Tony Paust. Transport to and from the Islands was provided by David Kingdom, Mal McRae, David van der Oord, Ian Rowlands, Chris Shine, Fred Tucker and Geoff Whitehurst. Thanks to Justin Parker, PeerJ Academic Editor Kenneth De Baets and two anonymous reviewers for improvements to the manuscript.

Additional Information and Declarations

Competing Interests

Author Contributions

1 Cellular reefs (terminology after Hoskin, 1963) are also widely known as reticulate reefs, from the latin reticulum: a network or net-like structure. However, the term reticulate has been applied to a variety of lagoon reef forms that probably develop through different mechanisms (Schlager & Purkis, 2015). Therefore we consider cellular reefs to be a subdivision of reticulate reefs, distinguished by subcircular depressions as shown in Fig. 1.

David Blakeway is an employee of Fathom 5 Marine Research.

David Blakeway conceived and designed the experiments, performed the experiments, analyzed the data, wrote the paper, prepared figures and/or tables.

Michael G. Hamblin conceived and designed the experiments, performed the experiments, reviewed drafts of the paper.

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
