# Peer review of "Self-generated morphology in lagoon reefs"

_PeerJ, doi:10.7717/peerj.935_

## Round 0.1 · original submission · Minor Revisions

Thank you for submitting this interesting work to PeerJ as attested by the positive and enthusiastic reviews on your approach as well as my own personal assessment. However, I concur with reviewer 2 that the structure of the manuscript including the abstract (see comments by reviewer 2) could be considerably improved. A more clear separation between methods, results and interpretations is necessary. I think this can be largely resolved in the text by adding a small methological section on the model which can be separate from the model results section.

Furthermore, several references are missing from the references list (Jones et al. 1994; Schlager and Purkis 2014; Purkis and Schlager 2014; Wyrwoll 2006; Delesalle 1985; Roth and Dehyn 2013) or the text (“Paulau-no reference yet”).

I also concur with reviewer 2, that you undersell your approach and results a little bit, which could be improved by writing some more lines and some more references on the importance of your novel approach for reef conservation, which can complement or precede traditional seismic, coring or remote sensing studies. Some more explanations on the difference between extrinsic and intrinsic factors (e.g., Aronson 2011) influencing reef growth are also in order. Adding a sentence to your abstract on these aspects might also considerable broaden its scope. More importantly, please clearly state how various types of reefs (nodular, cellular, etc.) and morphologies (egg carton) as well as coral morphologies (branching, etc.) are defined and list the references of the authors who have defined them in this way. All these points involve however rather minor changes, which can be quickly implemented.

In addition to the comments by the reviewers (particularly reviewer 2), the following points needs to be addressed in the revision:

Line 15-16: Please provide references for the following statements “Understanding … and predicting their future.”
Line 23: “Purkis and Schlager 2014” is missing from reference list
Line 25: Do you mean “MacNeill 1954” or “McNeill 1954” as it is listed in the reference list (please correct this in text or in reference list).
Line 35: “Schlager and Purkis 2014” is missing from the reference list, please add. Furthermore, could it be that you Schlager and Purkis 2013 as I could not find a paper by these authors (in this order) published in 2014
Line 35-36: maybe you should add directly in between “been” and “demonstrated”
Line 58: “Wyrwoll 2006” is missing from the references or do you mean “Wyrwoll et al. 2006”
Line 64: “Collins, Zhu & Wyrwoll” should be cited as “Collins et al.”; furthermore, “1996; 1998” is not properly formatting according to the PeerJ format
Line 119: “egg carton” form; is there not a more commonly used term for these morphology among coral workers; please define as it is not that clear from the provided illustrations/descriptions (see also comments of reviewer 2)
Line 145: please list some example taxa for “agariciids” and “pectiniids”; the reader might not be familiar with its composition, so it might also be helpful to cite the classification you are using here.
Line 187: “further back into the Holocene”; you need to be more specific why this is not possible: is it not preserved or can it not be reconstructed without seismic or coring studies
Line 191: “Model”; I guess calling this modelling results would be more appropriate here; a small description on methods before this would also be in order.
Line 199: “unconstrained but for sea level” is confusing, please replace by “unconstrained except for sealevel”
Line 331: “Wyrwoll 2006” is not listed in the reference list
Line 338: do you mean “Jones et al. 1994”, which is not listed in the references or rather “Jones et al. 1996”, which is listed in the references; please verify and correct
Line 349: please replace tsunami by “tsunamis”
Line 357: “Palau-no reference yet”: please add a reference or at least a personal communication or observation or delete Palau entirely.
Line 364: “Delesalle 1985” is missing from the reference list
Line 405: “invariably” is a bit strong as you are not sure in some cases: it would write “almost invariably” or “probably invariably” as this is closer to the current state of knowledge
Line 409: “Shinn 1976” is missing from the references
Line 420: “Roth and Dehyn 2013” are missing from the references
Line 426: “extrinsic influences”: please cite here and in other parts of the text or another reference to define the difference between extrinsic and intrinsic influences on reef growth => Aronson RB. 2011. Intrinsic And Extrinsic Drivers On Coral Reefs. Encyclopedia of Modern Coral Reefs: Structure, Form and Process:610-612.
Line 449-450: please list what would be necessary to recognize or study the formative Holocene or earlier state
Line 449-455: this approach could also be used before applying remote sensing or more expensive coring or seismic studies or to supplement these methods; please add something among these lines here as your approach really opens such possibilities
Line 735 – 768: all these references are out of place (not alphabetical); please rearrange and integrate them correctly (alphabetically) in other parts of the reference list

Reviewer 1 ·

Basic reporting

In my estimation this submission adheres to all PeerJ policies and demonstrates how the work fits into the broader field of knowledge with relevant prior literature properly referenced. I found the figures should be very good – clear and understandable. The article is nicely ‘self-contained,’ with a wealth of supplemental material available to those who want to go into a more detailed analysis or advance the model proposed by Blakeway and Hamblin. The research question is relevant and meaningful in my field of investigation. Conclusions are appropriately stated and based on solid observations and analysis. . I read through the manuscript and did not find any typographical errors. There writing is clear and concise. I did not check references, but these seem to be in order. It appears that setting up of the PeerJ reviewing pdf was done well.

Experimental design

I have 40 years of experience on coral reefs throughout the Pacific and I agree with their hypotheses and findings that shapes and forms of coral reefs represent antecedent geomorphology that is modified by biological response of corals and other calcifiers in a manner that can be viewed as a phenotypic expression of living reef-building organisms. For the most part reef ;morphology is generally attributed to external controls such as substrate topography or hydrodynamic influences. So the authors present an important question – how will reefs form in the absence of hydrodynamic gradients (within a calm lagoon) and pre-existing topography (flat Holocene foundation)? The Abrolhos cellular reefs provide an ideal site for study of cellular reef development due to a flat pre-Holocene substrate and very simple reef-building community consisting almost entirely of branching Acropora, with a few tabular Acropora in shallower water where there is more wave action. There modeling efforts demonstrate how features such as cellular and reticulate reefs can form purely as an expression of the growth and mechanical response of reef-building organisms (and bioeriosional process) and the resulting tendency of these structures to either remain in place or to break down.

Validity of the findings

As an observer of corals and coral reef development I have seen these processes in action, but this is all very complicated in most reef situations because of different antecedent platform conditions and variations in depth, current and wave action. So this paper is very valuable because it demonstrates what many of us have noted- as they say “reef morphology can be considered a phenotype of the predominant reef building organism”. One of the clearest examples is the widespread recognition by paleo ecologists is the analogy between micro atolls and reef formation. The conclusions are based on a very large and complete data set, and are statistically sound. The observations are made in a very uniform environment that is well-controlled. The conclusions follow well from the initial question, methods results and analysis.

Reviewer 2 ·

Basic reporting

In this manuscript the authors explore the intrinsic mechanisms that control reef morphology by using the reef growth simulations. By excluding external reef growth controls such as substrate and hydrodynamic from the reef growth simulations, the authors were able to identify morphology of reef-building organisms (branching or massive growth form) and their mechanical properties (fragility or sturdiness, respectively) as fundamental controls of overall reef morphology. It seems that reef morphology could be a proxy for the growth form of a most abundant reef-building organism, at least in the lagoon reefs.

The premise of the manuscript (to explore intrinsic controls of reef morphology) as well as the presented results (correlation between morphology or reef-building organisms and reef morphology) are both interesting and intriguing. The results satisfy basic scientific curiosity and if corroborated in different locations could potentially be used in conservation and remote sensing of reef systems. However, the manuscript needs major rewriting (rewording and restructuring) because it is really hard to follow. For example, authors do not use topic sentences for the paragraphs, they use colloquial (not clearly defined/described) phrases (such as ‘egg carton’ form), the text and the figures are not independent of each other.

Abstract:
- needs a topic sentence(s) that puts the study in a broader context.
- needs reorganisation. After a first two sentences that are background, there is a sentence that contains methods and results, followed by a sentence about the methods, followed by result sentence, followed by method sentence, and then followed by two result sentences.
- needs a conclusion/significance sentence.

Introduction:
- The first two-sentence paragraph is not introducing a topic properly and should be moved somewhere below
- The second paragraph seems more appropriate as an opening paragraph but it still needs expanding so both geological and biological aspects of reef morphology could be fully appreciated.
- Third sentence second paragraph: “But within the diversity is a subset of globally recurring forms, indicating that there are some universal rules governing reef morphology (Hatcher 1997; Blanchon 2011; Purkis and Schlager 2014).” What rules? Sentence needs expanding.
- First sentence, fifth paragraph: “While this work has removed all doubt…”. This work? This manuscript or one of four previously mentioned references? Is it possible to remove all doubt in science?
- Paragraph 6: “…seems better…”, “…arguably a much better…”. Are these authors personal assessments? What are these based on?
- Last paragraph introduces the study site, contains aim of the study and some background information on geological history of study site. However, the study site and some of it geological context was previously mentioned in the third paragraph.

Experimental design

There is no clear Methods section. The text (sections) between introduction and discussion needs restructuring. It contains information on methods, results and some study site introduction. I understand that the model was established by information learned from the field surveys and therefore the order of sections/paragraphs makes sense but the survey results should be better separated within the methods/results section.
One could maybe organise it in a following way:
1 Reef survey
1.1 Methods
1.2 Results
1.2.1 Circular reefs

2 Model
2.1. Methods
2.1.1 (Assumptions for) Configuration 1
2.1.2. (Assumptions for) Configuration 2
2.2 Results
2.2.1 Configuration 1


Survey methods:
- are missing some major information.
- “We quantified substrate composition by point counting video records of each transect, and constructed topographic profiles by recording tide-corrected depth every metre along each transect. We used twenty five categories in point count analysis and condensed these into seven categories for graphical representation: tabular Acropora, branching Acropora, other scleractinia, soft coral, macroalgae, dead coral, and sediment/rubble”. How many points per transect? How were there chosen? How long were the transects? What were the 25 categories? Maybe add a table with raw data in supplementary material.
- First sentence belongs to the introduction.

Enclosed depressions:
- “In the enclosed stage (sites C, D, I, O) the previously-established patterns strengthen…”. What previously-established patterns? Remind the reader, especially because this is a first sentence of a new subsection.

Model:
- second paragraph: “We used a consistent configuration for most model runs: an array of 160 x 160 cells (representing 160 x 160 m, or 2.56 hectares, of seafloor), a uniform depth of 30 m, and 64 randomly-spaced coral recruits, occupying 0.25% of the array (variations to this configuration are described later in supporting figures).” Why 160x160? Why 64 recruits, i.e. why 0.25%? The variations should be better discussed in the text.
- Methods and results of the model/simulations are mixed up and it is hard to follow.

Validity of the findings

Restructuring the introduction, methods and results section will help the reader to follow the discussion. To be able to better evaluate the results and discussion it is important to know what scleractinians belong to the category ‘other scleractinia’ and what were their growth forms.

Additional comments

Restructuring the manuscript and adding some additional information as suggested above will allow these interesting and important results to stand out more. Additionally the authors should highlight the broader (outside of the reef geomorphology field) importance of the main findings.

---

## Round 0.2 · Minor Revisions

Thanks for integrating all the suggestions and adding the missing references. Your manuscript is a good as accepted. Could you please still turn Fig. 4 to landscape as you suggested in your recent e-mail (such such changes cannot be made after (final) acceptance). Apart from this, no further revisions/changes are necessary.

Reviewer 2 ·

Basic reporting

Authors have adopted suggestions made during the first review. First of all, the manuscript structure has been notably improvement – there is a logical unfolding of information and the results clearly stand out. Moreover, vague statements and ones missing a scientific support have been adjusted, making the methods, results and discussion much stronger. Addition of supplementary material and rearrangement of figures has helped this cause. Finally, relating the results to a broader scientific context highlights the importance of research presented in this manuscript.

Experimental design

No Comments

Validity of the findings

No Comments

---

## Round 0.3 · accepted · Accept

Thanks for making these last changes.